# Prognostic significance of total choline on in-vivo proton MR spectroscopy for prediction of late recurrence in patients with hormone receptor-positive, HER2-negative early breast cancer

**Hyunjik Kim**[1], **Heungkyu Park**[1], **Yongsoon Chun**[1], **Hagjun Kim**[1], **Hyeonman Baek**[2], **Yunyeong Kim**[1]*

**1** Department of General Surgery, Breast Cancer Center, Gachon University Gil Medical Center, Incheon, Republic of Korea, **2** Department of Molecular Medicine, Lee Gil Ya Cancer and Diabetes Institute, Gachon University, Incheon, Republic of Korea

* crysblue511@gmail.com

**Data Availability Statement:** All relevant data are within the manuscript and its Supporting Information files.

## Abstract

### Purpose

In-vivo proton magnetic resonance spectroscopy (MRS) is a non-invasive method of analyzing choline metabolism that has been used to predict breast cancer prognosis. A strong choline peak may be a surrogate for aggressive tumor biology but its clinical relevance is unclear. The present study assessed whether total choline (tCho), as measured by proton MRS, can predict late recurrence in patients with hormone receptor (HR)-positive, HER2-negative early breast cancer.

### Methods

The study cohort included 261 HR+/HER2- breast cancer patients who underwent diagnostic single-voxel proton MRS (3.0T scanner) prior to first-line surgery from March 2011 to July 2014. The relationships between tCho compound peak integral (tChoi) values and others prognostic factor were analyzed, as were the effects of tChoi on 10-year disease-free survival (DFS) and overall survival (OS). The clinical significance of tChoi was also analyzed using Harrell's C-index.

### Results

Mean tChoi in HR+/HER2- study group was 15.47 and we set the cut-off for tChoi at 15 for survival analysis. 10-year DFS differed significantly between tChoi <15 and ≥15 (p = 0.017), with differences differing significantly for late (5–10 years; p = 0.02) but not early (0–5 years; p = 0.323) recurrence. Cox regression analysis showed that tChoi was significantly predictive of 10-year DFS (p = 0.046, OR 2.69) and tended to be predictive of late recurrence (HR 4.36, p = 0.066). Harrell's C-index showed that the Ki-67 index (AUC = 0.597) and

**Funding:** The author(s) received no specific funding for this work.

**Competing interests:** The authors have declared that no competing interests exist.

lymphovascular invasion (AUC = 0.545) were also predictive of survival, with the addition of normalized tChoi improving the AUC to 0.622 (p = 0.014), indicating better predictive power.

## Conclusion

tChoi determined by in vivo MRS was predictive of prognosis in patients with HR+/HER2- early breast cancer. This parameter may serve as a valuable, non-invasive tool to predict prognosis when combined with other known prognostic factors.

## Introduction

Breast cancer is the leading cause of death in women worldwide [1]. Although many treatment methods have been developed and survival rates have increased, breast cancer remains a complex and heterogeneous disease with various clinical challenges, especially in determining prognosis and assessing responses to treatment. Patients with luminal type breast cancer (hormone receptor positive-HER2/neu negative) have a good prognosis, with a relapse rate significantly lower than in patients with other breast cancer subtypes [2]. Nevertheless, recurrence remains inevitable in patients with luminal type breast cancer, making it difficult to predict both early (i.e., within 5 years after initial treatment) and late (i.e., more than 5 years after initial treatment) recurrence. A deeper understanding of the properties of individual tumors is therefore required to identify more precise prognostic factors.

In vivo proton magnetic resonance spectroscopy (MRS) measures shifts of particular nuclei in magnetic fields, allowing noninvasive molecular analysis [3]. The spectra produced by MRS represent all detectable metabolites with their individual chemical profiles in the region of interest [4]. The presence of a compound resonance around 3.23 ppm has been associated with several chemical compounds, including phosphoethanolamine, choline, phosphocholine, and glycerophosphocholine, with the latter three referred to as total choline (tCho) [5]. Increased tCho levels due to increased cellular membrane turnover have been observed in malignant tumors [6, 7], suggesting thatand these increasesd tCho may be an therefore serve as an indicator of malignancy [8]. Additionally, increased tCho levels have been associated with overexpression of the HER-2/neu gene [9] and with aggressive phenotypes, such as triple-negative breast cancer (TNBC) [10, 11]. MRS measurements of tCho levels have also been used to assess axillary lymph node metastases [12], to evaluate responses to neoadjuvant chemotherapy or radiation therapy [13, 14], and to evaluate the relationship of tCho with pathologic prognostic factors in primary breast cancer [12, 15, 16]. Despite these findings, however, in vivo proton MRS remains inadequate for clinical applications, and related further research has been limited.

A recent study of the relationship between ERα and choline metabolism found that ERα directly regulated the gene encoding Cho phosphotransferase 1 (CHPT1), an enzyme necessary for estrogen to affect Cho metabolism, including increased phosphatidylcholine synthesis [17]. This finding suggests that choline metabolism is involved in the development and/or progression of hormone receptor (HR)-positive, relatively non-aggressive breast cancer, as opposed to the more aggressive subtypes of breast cancer studied previously.

The present study therefore evaluated whether tCho, as measured by in vivo proton MRS, could predict 10-year survival, in particular late recurrence, in patients with HR-positive, HER2-negative early breast cancer.

## Methods

### MRI and in vivo proton MR spectroscopy (single-voxel)

Single-voxel proton MRS was performed using a 3.0-T MRI scanner (Skyra and Verio, Siemens Medical Solutions, Erlangen, Germany) with a dedicated bilateral receiver-only phased-array four-element two-channel coil (one channel per breast) The protocol for bilateral breast imaging consisted of an axial STIR sequence (TR/TE, 7820/86; inversion time, 120 ms; 3-mm thickness without an interslice gap; FOV, 180 × 180 mm2; matrix size, 512 × 512; scan time; 1–2 min), a 3D T1-weighted FLASH dynamic gradient-echo sequence (TR/TE, 4.5/1.6; flip angle, 10˚; 0.9-mm thickness without an interslice gap; 0.9 × 0.9 × 0.9 mm3 isotropic voxel; one unenhanced and four contrast-enhanced acquisitions with a temporal resolution of 192 seconds), and, for dynamic contrast enhancement, the injection of 0.1 mmol/kg body weight gadobutrol (Gadovist; Schering AG, Berlin, Germany), followed by a 20-mL saline flush and an axial 3D delayed contrast-enhanced turbo spin-echo pulse sequence (TR/TE, 5.6/2.5; FOV, 380 × 380 mm2; matrix size, 384 × 384; slice thickness, 1.5 mm) to evaluate supraclavicular and axillary lymph nodes.

Lesion size was determined by measuring the largest diameter of those obtained from the first or second subtracted axial images and their sagittal and coronal reconstructions.

The technical parameters for the proton MRS sequence consisted of TR/TE, 7.6/3.53; flip angle, 20˚; and spectral width, 890Hz. The total time to acquire MRS per lesion, including scan time and shimming, usually ranged from 8 to 10 minutes. MRS row data were post-processed on a remote workstation using software supplied by the manufacturer (SW Numaris 4; Siemens Healthcare). For the water- and fat-suppressed spectra used to measure the tCho-containing compound peak, postprocessing was systematically performed to correct every signal with the zero-order phase of its residual water peak after Fourier transformation. Baseline corrections were performed to exclude ranges for lipids (0–2.8 ppm), water (4.0–6.0 ppm), and tCho (3.18–3.28 ppm). Curve fitting using a Gaussian function in the range of tCho (3.18–3.28 ppm) was finally applied to calculate the tCho peak integral. The tCho containing compound peak integral (tChoi) was expressed in arbitrary units (AU). Baseline oscillations with multiple peaks between 3.18 and 3.28 ppm, each with an integral lower than 0.1 arbitrary units, were excluded from the study as measurement errors.

The volume of interest (VOI) was a rectangular box, positioned by a radiologist with 10 years of experience in breast MRI based on axial, coronal, and sagittal subtraction images. The VOI positions were chosen to be within each enhancing lesion as much as possible, with the goal of minimizing the inclusion of nonenhancing glandular tissue or surrounding fat. VOI size was defined as a single voxel of 1.0 X 1.0 X 1.0 cm, equivalent to 1 mL in volume, with the tChoi normalized by dividing by the VOI (AU/1 mL).

Example of spectra obtained by MRS and patient data are shown in Fig 1.

### Study variables and pathologic analysis

Tumors were staged pathologically in accordance with the American Joint Committee on Cancer (AJCC) staging system, with all tumors reclassified based on the TNM 8th edition. Pathological and MRI tumor sizes were compared using a 2-cm threshold, with lymph node metastasis classified as positive (N1 or N2) or negative (N0).

The levels of expression of estrogen receptor (ER) and/or progesterone receptor (PR) were scored immunohistochemically (IHC) using the Allred score (AS), with a maximum score of 8, and an AS score ≥3 considered positive. HER2/neu status was measured using IHC and

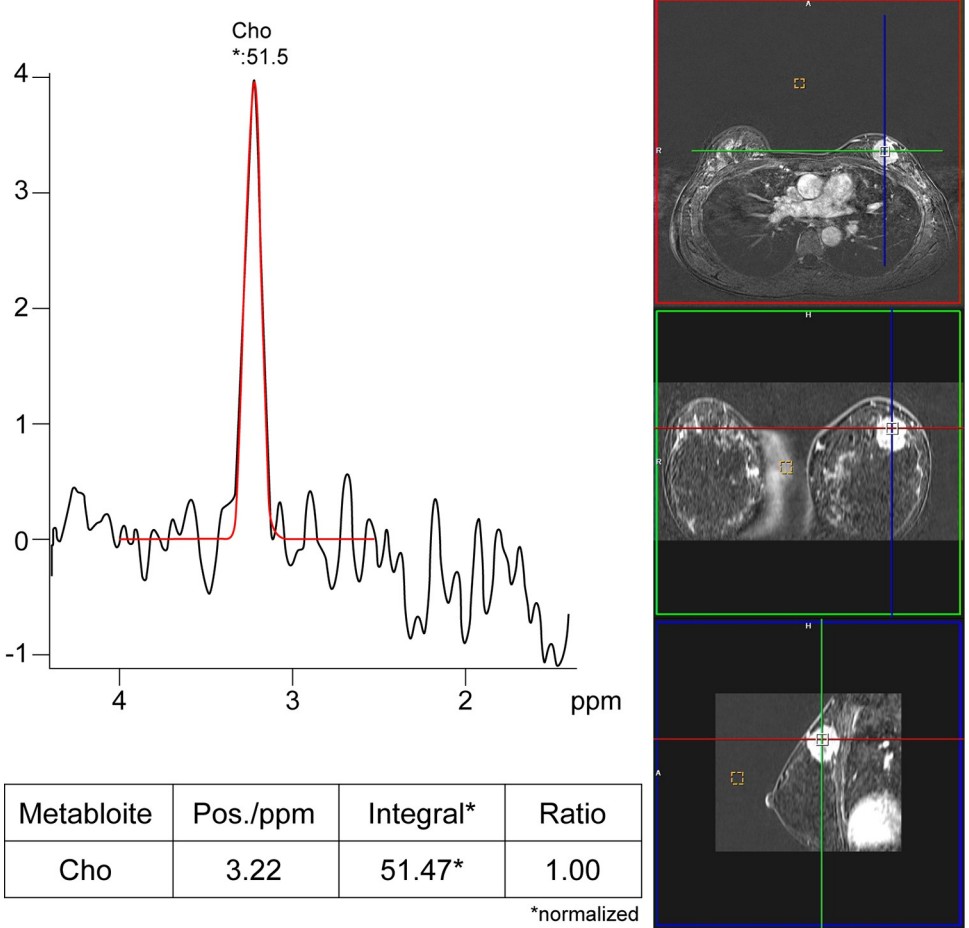

| Metabloite | Pos./ppm | Integral* | Ratio |
|---|---|---|---|
| Cho | 3.22 | 51.47* | 1.00 |

*normalized

**Fig 1. Single voxel MR spectroscopy images of a 42-year-old woman, showing an enhancing lesion about 2.8 cm in diameter with irregular margins at the upper center of the left breast.** The position of the volume of interest (VOI), measuring 1.0 X 1.0 X 1.0 cm, is shown on the right for each of the three orthogonal reference planes. The magnified proton MR spectrum (range, 0.00–6.00 ppm) on the right shows the total choline-containing compound peak at 3.22 ppm. The total choline-containing compound peak integral was 51.47 arbitrary units (AU) and was normalized by dividing by 1 mL (single voxel).

silver-enhanced in situ hybridization (SISH). HER2/neu positivity was defined as an intensity of 3+ by IHC or 2+ by SISH.

Other prognostic factors analyzed included age, Ki-67 index, lymphovascular invasion (LVI), histologic grade, nuclear grade, extensive intraductal component (EIC), and clinical risk score. Age ≤50 years was classified as high risk, and age >50 years as low risk. The cutoff for the Ki-67 index was set at 20%, which can distinguish between Luminal A and Luminal B breast cancer subtypes. Tumor grade (histologic or nuclear) was determined using the Nottingham method (with a range of 1 to 3), with grade 3 considered high risk, and grades 1 and 2 considered low risk. EIC was considered predictive of local recurrence after breast-conserving surgery and radiotherapy. Ductal carcinoma in situ (DCIS) occupying >25% of the cancer area or extending beyond the edge of the cancer was classified as tumorous. Clinical risk score was determined using the modified Adjuvant! Online tool, as reported in the MINDACT trial [18].

**Table 1. Total choline containing compound integral (tChoi) in the study population.**

| Total choline containing compound integral (tChoi) | | Value |
|---|---|---|
| tChoi (AU/1mL), mean (range) | | 15.47 (0.13–55.70) |
| tChoi with mean value cutoff | (n, %) | |
| | <15 | 158 (60.5%) |
| | ≥15 | 103 (39.5%) |

## Statistical analysis

The cutoff for tChoi was set at 15, with the mean tCho represented as a continuous variable in patients with HR+/HER2- breast cancer. This cutoff demonstrated significance in the subsequent Kaplan-Meier analysis and was therefore utilized in subsequent multivariable survival analyses (Table 1).

Normally distributed continuous variables, as determined by Kolmogorov-Smirnov tests, were presented as mean and standard deviation, whereas categorical variables were presented as frequencies and percentages. Because the Kolmogorov-Smirnov test for normality indicated that tumor size, as determined by pathology and MRI; age; and tChoi were normally distributed ($p < 0.05$ each), they were compared by independent sample t-tests. 10-year disease-free survival (DFS) and overall survival (OS) were analyzed by the Kaplan-Meier method, with subgroups compared by log-rank test. Multivariable Cox regression analysis was performed to determine factors significantly associated with patient survival, with Harrell's C-index used to evaluate tChoi as a prognostic factor in combination with other known prognostic factors. All statistical analyses were performed using SPSS for Windows (Version 23.0), with all tests being two-tailed and p-values $< 0.05$ considered statistically significant.

## Ethical considerations

This study involved human participants as well as human data or tissues, with a focus on patients undergoing surgical treatment for breast cancer. On the day prior to surgery, as part of the process of signing the surgical consent form, the medical institution requires all patients to also provide consent for the donation and research use of human-derived materials. There were no minor participants, and informed consent was obtained from all participants. The recruitment period was from March 1, 2011 to July 30, 2014 and the access for research purposes was on August 1, 2023. The authors have access to the data of individual participants who have been anonymized. The study was approved by the institutional review board of Gachon University Gil Medical Center (GMC) (IRB No. GAIRB2023-267).

## Results

### Study population

This was a single-center retrospective cohort study, in which all breast cancer patients from March 2011 to July 2014 underwent initial MRI along with in vivo proton MRS for research and diagnostic purposes. Of the 815 patients with MRS measurements, 148 were excluded based on exclusion criteria and an additional 107 were excluded due to criteria for rejecting spectra, which involved the presence of artifacts in the region of metabolites containing tCho compounds (range, 2.8–4.0 ppm) or failure to suppress water and fat signals. Of the 560 early breast cancer patients who underwent in vivo proton MRS, the study group (HR+/HER2-) consisted of 261 patients (Fig 2).

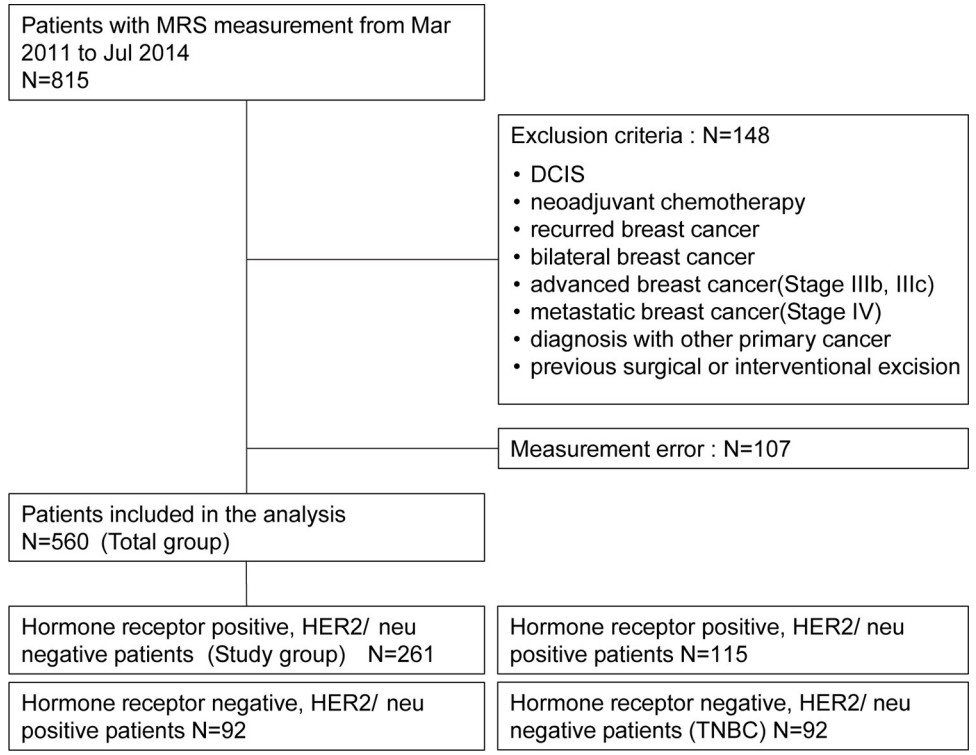

**Fig 2. Study flow chart.**

## Clinical and pathologic characteristics

Of the 261 HR+/HER2 patients, 54.8% and 45.3% had Ki-67 index <20 and >20, respectively, and 62.1% were negative for lymphovascular invasion. Of these patients, 54.8% were histologic grade II and 67.4% were nuclear grade II, with 71.3% being negative for EIC. Assessment of clinical risk showed that 47.9% of these patients were low risk and 52.1% were high risk (Table 2).

## Comparison of total choline

The mean tChoi in study group was 15.47(range, 0.13–55.7). Based on the mean tChoi in the HR+/HER2- study group, the cut-off for tChoi was set at 15 for subsequent survival analyses (Table 1).

Although mean tChoi was not significantly associated with other prognostic factors in these patients, higher mean tChoi tended to be associated with higher histologic grade (14.83 vs 17.93, p = 0.115), nuclear grade (14.82 vs 18.02, p = 0.091), and clinical risk (14.43 vs 16.42, p = 0.207) in the high-risk group (Table 3).

## Survival analysis–Univariable vs Multivariable

Kaplan-Meier analysis of patients with HR+/HER2- breast cancer showed that 10-year DFS differed significantly between patients with tChoi <15 and ≥15, with mean DFS times of 119.11 months and 115.09 months, respectively (log rank p = 0.017). The p-value for early recurrence (0–5 years) was 0.323, whereas the p-value for late recurrence (6–10 years) was 0.020. 10-year OS rates did not differ significantly between HR+/HER2- breast cancer patients with tChoi <15 and ≥15 (Tables 4 and 5 and Fig 3). Tables 4 and 5

**Table 2. Clinical and pathologic characteristics of patients in the study population.**

| Characteristics | | Value |
|---|---|---|
| **n** | | 261 |
| **Age (y), mean (range)** | | 50.6 (31–82) |
| **Tumor size at pathology(cm), mean(range)** | | 2.2 (0.2–9) |
| **Tumor size on MRI (cm), mean (range)** | | 2.1 (0.6–7.5) |
| **Pathologic T stage** | (n, %) | |
| | T1mi, T1 | 141 (54%) |
| | T2 | 105 (40.2%) |
| | T3 | 15 (5.7%) |
| **Pathologic N stage** | (n, %) | |
| | N0, N1mi | 165 (64.8%) |
| | N1 | 75 (28.7%) |
| | N2 | 17 (6.5%) |
| **Pathologic Stage** | (n, %) | |
| | I | 110 (42.1%) |
| | II | 126 (48.3%) |
| | IIIA | 25 (9.6%) |
| **Histologic subtype** | (n, %) | |
| | Ductal | 222 |
| | Lobular | 20 |
| | Mixed | 2 |
| | Micropapillary | 5 |
| | Metaplastic | 0 |
| | Favorable (other) | 12 |
| **ER** | (n, %) | |
| | Negative | 6 (2.3%) |
| | Positive | 255 (97.7%) |
| **PR** | (n, %) | |
| | Negative | 31 (11.9%) |
| | Positive | 230(88.1%) |
| **HER2** | (n, %) | |
| | Negative | 261 (100%) |
| | Positive | 0 (0%) |
| **Ki-67 index** | (n, %) | |
| | <20 | 114 (54.8%) |
| | ≥20 | 118 (45.2%) |
| **Lymphovascular invasion** | (n, %) | |
| | Negative | 162 (62.1%) |
| | Positive | 99 (37.9%) |
| **Histologic grade** | (n, %) | |
| | I | 64 (24.5%) |
| | II | 143 (54.8%) |
| | III | 54 (20.7%) |
| **Nuclear grade** | (n, %) | |
| | I | 32 (12.3%) |
| | II | 176 (67.4%) |
| | III | 53 (20.3%) |
| **Extensive intraductal component (EIC)** | (n, %) | |

*(Continued)*

**Table 2.** (Continued)

| Characteristics | | Value |
|---|---|---|
| | Negative | 186 (71.3%) |
| | Positive | 75 (28.7%) |
| **Clinical risk (Adjuvant! Online)** | (n, %) | |
| | Low risk | 125 (47.9%) |
| | High risk | 136 (52.1%) |

Note–Favorable (other) = pure mucious, pure cribriform, solid papillary carcinoma (SPC), total 12 patient.

**Table 3. Relationship of total choline containing compound integral (tChoi) with other prognostic factors.**

| Prognostic value | | tChoi(AU/1mL) / mean (SD) |
|---|---|---|
| **Age, years** | | |
| | ≥50 | 15.43(13.08) |
| | <50 | 15.50(12.40) |
| | p value | 0.967 |
| **Pathologic T stage** | | |
| | T1mi/T1 (<2cm) | 15.83(13.57) |
| | T2/T3 (≥2cm) | 15.04(11.63) |
| | p value | 0.615 |
| **MRI size** | | |
| | <2cm | 15.50(13.34) |
| | ≥2cm | 15.43(11.95) |
| | p value | 0.968 |
| **Lymph node metastasis** | | |
| | negative | 15.49(13.35) |
| | positive | 15.43(11.47) |
| | p value | 0.967 |
| **Pathologic stage** | | |
| | I | 15.83(13.57) |
| | II | 15.06(11.76) |
| | IIIA | 14.89(11.10) |
| | p value | 0.880 |
| **ER** | | |
| | negative | 16.64(9.22) |
| | positive | 15.44(12.78) |
| | p value | 0.766 |
| **PR** | | |
| | negative | 15.64(13.69) |
| | positive | 15.44(12.59) |
| | p value | 0.940 |
| **Ki-67 index** | | |
| | <20 | 15.60(13.64) |
| | ≥20 | 15.30(11.51) |
| | p value | 0.848 |
| **Lymphovascular invasion** | | |
| | negative | 15.26(13.20) |

(*Continued*)

**Table 3.** (Continued)

| Prognostic value | | tChoi(AU/1mL) / mean (SD) |
|---|---|---|
| | positive | 15.80(11.89) |
| | p value | 0.728 |
| **Histologic grade** | | |
| | Low (I / II) | 14.83(12.63) |
| | High (III) | 17.93(12.78) |
| | p value | 0.115 |
| **Nuclear grade** | | |
| | Low (I / II) | 14.82(12.82) |
| | High (III) | 18.02(11.99) |
| | p value | 0.091 |
| **Extensive intraductal component (EIC)** | | |
| | negative | 16.14(12.54) |
| | positive | 13.81(13.02) |
| | p value | 0.190 |
| **Clinical risk (Adjuvant! Online)** | | |
| | low risk | 14.43(13.21) |
| | high risk | 16.42(12.18) |
| | p value | 0.207 |

**Table 4. Kaplan-Meier analyses of 10-year disease-free survival (DFS) and overall survival (OS) in patients with HR+/HER2- breast cancer.**

| | 10Y-DFS | 10Y-OS |
|---|---|---|
| **tChoi<15** | 119.11 (116.81–121.42) | 121.81 (121.44–122.18) |
| **tChoi≥15** | 115.09 (110.64–119.53) | 120.91 (119.73–122.09) |
| **p value(log-rank)** | 0.017 | 0.111 |

**Table 5. Kaplan-Meier analyses of 0–5 year (early recurrence) and 5–10 year (late recurrence) disease-free survival (DFS) in patients with HR+/HER2- breast cancer.**

| | | tChoi (AU/1mL) | DFS | recurrence (n) | p value(log-rank) |
|---|---|---|---|---|---|
| **0–5 years (early recurrence)** | tChoi<15 | 8.23 (4.06–13.8) | 44.5 (34–60) | 4 | |
| | tChoi≥15 | 36.7 (20.3–53.9) | 26.4 (13–43) | 5 | 0.323 |
| **5–10 years (late recurrence)** | tChoi<15 | 5.02 (3.01–7.03) | 79.5 (79–80) | 2 | |
| | tChoi≥15 | 23.72 (15.4–38.64) | 101 (69–121) | 8 | 0.020 |

Multivariable Cox proportional hazard regression analysis showed that Ki-67 index (HR 3.28, 95% CI 1.04–10.32, p = 0.042), lymphovascular invasion (HR 2.97, 95% CI 1.03–8.6, p = 0.044), and tChoi (HR 2.69, 95% CI 1.02–7.09, p = 0.046) were significantly predictive of 10-year DFS in patients with HR+/HER2- breast cancer. Histologic grade tended to be associated with early (0–5 years) recurrence (HR 6.07, p = 0.07), whereas age (HR = 0.28, p = 0.07) and tChoi (HR 4.36, p = 0.066) tended to be associated with late (5–10 years) recurrence (Table 6).

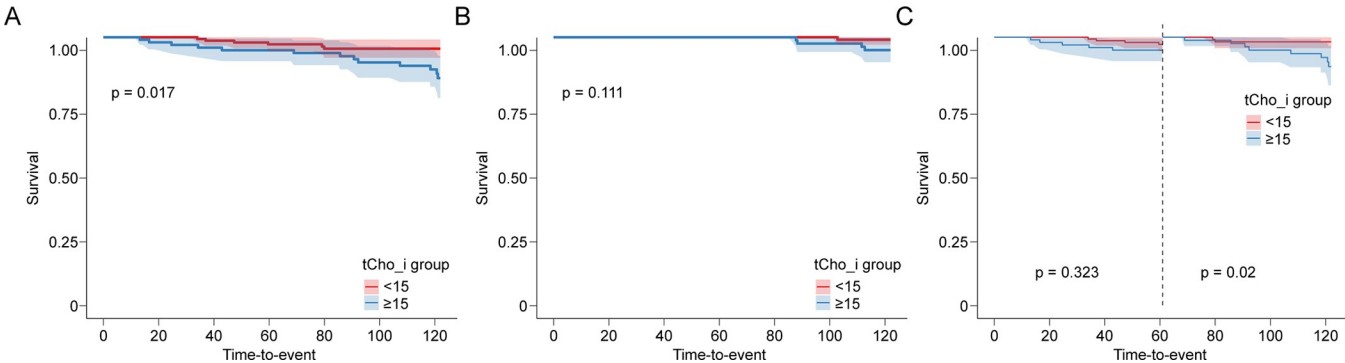

**Fig 3. Kaplan-Meier survival curves for 10-year disease free survival (DFS) and overall survival (OS).** (A) 10-year DFS in the HR+/HER2- study group. (B) 10-year OS in the HR+/HER2- study group. (C) 0–5 year (early recurrence) and 5–10 year (late recurrence) DFS in the HR+/HER2- study group.

The clinical significance of tChoi was additionally analyzed using Harrell's C-index to compare areas under receiver operating characteristic (ROC) curves (AUCs). When tChoi was analyzed independently, it yielded an AUC of 0.506 (95% CI 0.439–0.573). The predictive accuracy of tChoi was improved when assessed in conjunction with lymphovascular invasion (AUC 0.553) and Ki-67 index (AUC 0.606), with the three factors having an AUC of 0.622 (95% CI 0.555–0.689). The difference in AUC between tChoi alone and the combination of the three variables was found to be statistically significant (p = 0.014), indicating that incorporating lymphovascular invasion and Ki-67 index with tChoi resulted in improved predictive ability (Table 7).

## Discussion

Although mean tChoi was not significantly associated with other prognostic factors in the present study, mean tChoi tended to be higher in more aggressive tumor (Table 3). Similarly,

**Table 6. Multivariable Cox proportional hazard regression analysis of factors associated with survival analysis in patients with HR+/HER2- breast cancer.**

| | 0–5 years (early recurrence) DFS | | 5–10 years (late recurrence) DFS | | 10-yr DFS | |
|---|---|---|---|---|---|---|
| **Variable** | HR(95% CI) | p value | HR(95% CI) | p value | HR(95% CI) | p value |
| **Age group:**<br>**<50 vs ≥50 years** | 4.71 (0.56,39.41) | 0.15 | 0.28 (0.07,1.12) | 0.07 | 0.88 (0.34,2.27) | 0.79 |
| **Pathologic T stage:**<br>**ref. = T1** | | | | | | |
| **T2** | 1.04 (0.19,5.64) | 0.964 | 1.56 (0.36,6.8) | 0.557 | 1.3 (0.43,3.91) | 0.642 |
| **T3** | N/A | 1 | 2.04 (0.17,25.05) | 0.579 | 0.86 (0.09,8.56) | 0.898 |
| **Pathologic N stage:**<br>**ref. = N0, N1mi** | | | | | | |
| **N1** | 0.35 (0.06,2.12) | 0.251 | 2.95 (0.61,14.17) | 0.177 | 1.06 (0.34,3.26) | 0.924 |
| **N2** | 1.61 (0.25,10.44) | 0.616 | 2.1 (0.15,28.63) | 0.579 | 2 (0.44,9.08) | 0.371 |
| **ki-67 index:**<br>**≥20 vs <20** | 3.76 (0.38,37.46) | 0.259 | 2.87 (0.61,13.55) | 0.182 | 3.28 (1.04,10.32) | 0.042 |
| **Lymphovascular invasion: positive vs negative** | 2.78 (0.43,17.74) | 0.28 | 1.7 (0.34,8.62) | 0.519 | 2.97 (1.03,8.6) | 0.044 |
| **Histologic grade:**<br>**high(III) vs low(I/II)** | 6.07 (0.86,42.82) | 0.07 | 0.78 (0.06,10.68) | 0.851 | 2.93 (0.71,12.12) | 0.138 |
| **Nuclear grade:**<br>**high(III) vs low(I/II)** | 0.38 (0.06,2.35) | 0.296 | 0.91 (0.06,12.95) | 0.944 | 0.4 (0.1,1.66) | 0.209 |
| **tChoi: ≥15 vs <15** | 2.49 (0.56,11.13) | 0.232 | 4.36 (0.91,20.88) | 0.066 | 2.69 (1.02,7.09) | 0.046 |

Note–ref. = reference, DFS = disease free survival, HR = hazard ratio, tChoi = total choline containing compound integral.

**Table 7.  Harrell's C-index for tChoi and its combination with each prognostic factor, as determined by differences in AUC.**

| Harrell's C-index in each prognostic factor combination | | | | | | | | | |
|---|---|---|---|---|---|---|---|---|---|
| Each factor | | | Combination of 2 factors | | | Combination of 3 factors | | | |
| | AUC | 95% CI | | AUC | 95% CI | | AUC | 95% CI | P-value for AUC, difference with tChoi only |
| tChoi (cut off 15) | 0.506 | 0.439–0.573 | lym.inv +tChoi | 0.553 | 0.486–0.62 | Ki-67+lym.inv +tChoi | 0.622 | 0.555–0.689 | 0.014 |
| Lymphovascular invasion | 0.545 | 0.478–0.612 | Ki-67+tChoi | 0.606 | 0.538–0.673 | | | | |
| Ki-67 (cut off 20%) | 0.597 | 0.53–0.664 | | | | | | | |

higher phosphocholine (Pcho) levels have been reported to be associated with more aggressive histologic grade 3 tumors [19]. Increased levels of Pcho in breast cancer cells have been attributed to the oncogenic activity of choline kinase (CK), the enzyme that converts choline into Pcho, with CK activity also reported to be strongly associated with high histologic grade [20].

The present study also found that tChoi levels were significantly associated with 10-year DFS, particularly late recurrence, in patients with HR+/HER2- tumors. Moreover, the predictive power of tChoi was enhanced when combined with other factors, including lymphovascular invasion and Ki-67 index. Previous studies have also reported an association between CHPT1 and ERα. For example, the CHPT1 gene, which is directly regulated by ERα, was required for estrogen-induced effects on Cho metabolism, including increased phosphatidyl-choline (PtdCho) synthesis [17]. Additionally, immunohistochemical (IHC) analysis showed that the expression of CHPT1 is higher in breast cancer cells than in normal breast tissue and is higher in ER-positive than in ER-negative breast cancer [17]. The present study showed that survival rates following late recurrence were associated with tCho levels. CHPT1 also plays a role in the invasion of tamoxifen-resistant breast cancer cells, with tamoxifen-resistant LCC2 cells exhibiting greater invasiveness than tamoxifen-sensitive MCF7 cells under CHPT1 expression [17]. Moreover, CHPT1 depletion resulted in stronger suppression of invasion and metastasis by LCC2 cells than by MCF7 cells [17]. ER-positive breast cancer has often been associated with late recurrence [21], with extended hormone therapy administered to many patients with ER-positive breast cancer to reduce the risk of recurrence [22]. Adherence to hormone therapy, however, often decreases after 5 years [23], with many patients not receiving extended hormone therapy. This environment of minimal residual disease (MRD) may function similarly to tamoxifen-resistant breast cancer cells, potentially increasing CHPT1 activity.

The present study confirmed that levels of tCho are higher in more aggressive breast cancer phenotypes, including tumors with HER-2/neu overexpression and TNBC [9–11], with similar findings observed in HR+/HER2- luminal-type breast cancer. Although studies investigating choline metabolism in breast cancer and agents targeting these pathways are ongoing, comprehensive biological evidence for the underlying mechanisms is still lacking, indicating the need for additional research.

This study had several limitations. First, its design was retrospective, although patients were enrolled consecutively. The follow-up observation period has been set by our medical center at 10 years; however, due to the long study duration, many patients have been lost to follow-up. Furthermore, treatment approaches may have differed, due to changes in guidelines and variations in patient compliance. Large prospective studies are therefore required to confirm these results. Second, the tChoi of the total mass was not quantified. Several methods are available to quantitate tCho, including absolute tChoi, SNR, and internal/external reference approaches [24–26]. The latter method requires additional time to collect data from an internal or external

reference, to correct partial volume effects, and to carefully calibrate differences in relaxation times between the tissue tCho-containing compound signal and references [25]. The present study focused on the qualitative analysis of tCho, after positioning a single-sized voxel within each tumor, and aimed to determine an appropriate cutoff value to analyze differences in survival. However, the absolute quantification of tCho concentrations may be more desirable to assess cancer lesions. Third, the present study included tumors <1 cm in diameter. A single VOI measuring 1.0 x 1.0 x 1.0 cm was positioned, regardless of tumor size, and tCho was measured. Measurements may therefore be lower than the actual values for tumors <1 cm in size during postprocessing, especially water-fat suppression due to the increased proportion of water and fat around the tumor. In some patients, however, an ideal choline peak was detected even in smaller-sized tumors; because these tumors were <10% of the total, they were included in the study results, which may have affected the overall outcomes. Fourth, this study also included patients with multifocal/multicentric breast cancer. In these patients, the tCho of the largest main lesion was measured, although choline specificity may vary among individual tumors. This variation may have affected the study results.

## Conclusion

MRS parameters may play a role as biomarkers for predicting late recurrence of HR+/HER2- early breast cancer. AUC analysis showed that tChoi had a greater predictive ability when combined with previously identified prognostic factors than when considered alone. Although further prospective studies in larger patient cohorts are necessary, tChoi measured by in vivo MRS can serve as a valuable and non-invasive tool to predict prognosis when combined with other established prognostic factors.

## Author Contributions

**Conceptualization:** Hyunjik Kim, Yunyeong Kim.

**Data curation:** Hyunjik Kim, Yunyeong Kim.

**Formal analysis:** Hyunjik Kim, Yunyeong Kim.

**Funding acquisition:** Yunyeong Kim.

**Investigation:** Hyunjik Kim, Heungkyu Park, Yongsoon Chun, Hagjun Kim.

**Methodology:** Hyunjik Kim, Yunyeong Kim.

**Project administration:** Yunyeong Kim.

**Resources:** Heungkyu Park, Yunyeong Kim.

**Software:** Hyunjik Kim, Hyeonman Baek, Yunyeong Kim.

**Supervision:** Hyunjik Kim, Yunyeong Kim.

**Validation:** Hyeonman Baek, Yunyeong Kim.

**Visualization:** Hyunjik Kim, Yongsoon Chun.

**Writing – original draft:** Hyunjik Kim.

**Writing – review & editing:** Yunyeong Kim.

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
