## [Decision Letter · Decision Letter 0]

17 Nov 2024

PONE-D-24-39699Prognostic significance of total choline on in-vivo proton MR spectroscopy for prediction of late recurrence in patients with hormone receptor-positive, HER2-negative early breast cancerPLOS ONE

Dear Dr. Kim,

Thank you for submitting your manuscript to PLOS ONE. After careful consideration, we feel that it has merit but does not fully meet PLOS ONE’s publication criteria as it currently stands. Therefore, we invite you to submit a revised version of the manuscript that addresses the points raised during the review process.

We look forward to receiving your revised manuscript.

Kind regards,

Seok-Geun Lee, PhD

Academic Editor

PLOS ONE

Journal Requirements:

Reviewers' comments:

Reviewer's Responses to Questions

**Comments to the Author**

1. Is the manuscript technically sound, and do the data support the conclusions?

Reviewer #1: Yes

Reviewer #2: Yes

2. Has the statistical analysis been performed appropriately and rigorously? 

Reviewer #1: Yes

Reviewer #2: Yes

3. Have the authors made all data underlying the findings in their manuscript fully available?

Reviewer #1: Yes

Reviewer #2: Yes

4. Is the manuscript presented in an intelligible fashion and written in standard English?

Reviewer #1: Yes

Reviewer #2: Yes

5. Review Comments to the Author

Reviewer #1: This is a well written paper exploiting tCho as a biomarker to predict DFS and OS. I believe it is worth publishing.

My comments are written here below:

- line 134: authors should make clear that the Ki67% threshold of 20% is defining luminal A from luminal B tumors.

- line 178: if this was a retrospective study,it means that tCho was part of the routine of breast MRI imaging which is not the case in most institutions. Are all patients receiving pre operative breast MRI in this center, also receive MRS? can we have more explanation concerning this institutional policy?

Reviewer #2: This manuscript addresses a significant and clinically relevant question by investigating the role of total choline (tCho) measured by in vivo proton magnetic resonance spectroscopy (MRS) as a prognostic biomarker for hormone receptor-positive (HR+)/HER2-negative early breast cancer. The authors successfully demonstrate the association between elevated tCho and long-term outcomes, particularly late recurrence (5–10 years), and propose tCho as a potential non-invasive biomarker for clinical applications. The study design, methodology, and analysis are robust, and the results are clearly presented.

While the study is robust, the addition of expanded biological context, more detailed discussion of clinical applications would enhance its overall impact.

Reviewer’s suggestions

1. Introduction

Add a brief mention of CHPT1 and its role in choline metabolism to set up the biological context for the results.

2. Results

Table 3 shows the relationship between total choline (tCho) and prognostic factors like histologic and nuclear grades. Besides highlighting statistically significant results (p < 0.05), it is valuable to discuss findings with clinically meaningful trends, even if they lack statistical significance. Including these insights can enhance the analysis and improve readers' understanding of the data.

3. Discussion

The manuscript highlights a stronger association between tCho and late recurrence than early recurrence. An analysis of the potential reasons for this difference would enhance the interpretation of the findings.

6. PLOS authors have the option to publish the peer review history of their article (what does this mean?). If published, this will include your full peer review and any attached files.

Reviewer #1: **Yes: **Stylianos Drisis

Reviewer #2: No

---

## [Author Response · Author response to Decision Letter 0]

6 Dec 2024

Reviewer #1

Comment 1:

-line 134: authors should make clear that the Ki67% threshold of 20% is defining luminal A from luminal B tumors.

Reply 1: We clarify it on manuscript in Study variables and pathologic analysis park, line 141-143 

The cutoff for the Ki-67 index was set at 20%, which can distinguish between Luminal A and Luminal B breast cancer subtypes.

Comment 2:

-line 178: if this was a retrospective study,it means that tCho was part of the routine of breast MRI imaging which is not the case in most institutions. Are all patients receiving pre operative breast MRI in this center, also receive MRS? can we have more explanation concerning this institutional policy?

Reply 2: We clarify it on manuscript in Study population part, line 184-186 

This was a single-center retrospective cohort study, in which all breast cancer patients from March 2011 to July 2014 underwent initial MRI along with in vivo proton MRS for research and diagnostic purposes.

Reviewer #2

Comment 1:

1. Introduction

Add a brief mention of CHPT1 and its role in choline metabolism to set up the biological context for the results.

Reply 1: I have inserted the relevant content into the Introduction section. Line 73-78

A recent study of the relationship between ERα and choline metabolism found that ERα directly regulated the gene encoding Cho phosphotransferase 1 (CHPT1), an enzyme necessary for estrogen to affect Cho metabolism, including increased phosphatidylcholine synthesis [17]. This finding suggests that choline metabolism is involved in the development and/or progression of hormone receptor (HR)-positive, relatively non-aggressive breast cancer, as opposed to the more aggressive subtypes of breast cancer studied previously.

Comment 2:

2. Results

Table 3 shows the relationship between total choline (tCho) and prognostic factors like histologic and nuclear grades. Besides highlighting statistically significant results (p < 0.05), it is valuable to discuss findings with clinically meaningful trends, even if they lack statistical significance. Including these insights can enhance the analysis and improve readers' understanding of the data.

Reply 2: The mentioned content was written in the Results and Discussion sections of the previous draft. Line 210-213(result), Line 262-268(Discussion) 

Comment 3:

3. Discussion

The manuscript highlights a stronger association between tCho and late recurrence than early recurrence. An analysis of the potential reasons for this difference would enhance the interpretation of the findings.

Reply 3: I have added and revised a paragraph discussing the association between CHPT1, tamoxifen-resistant cells, and late recurrence.

Previous studies have also reported an association between CHPT1 and ERα. For example, the CHPT1 gene, which is directly regulated by ERα, was required for estrogen-induced effects on Cho metabolism, including increased phosphatidylcholine (PtdCho) synthesis [17]. Additionally, immunohistochemical (IHC) analysis showed that the expression of CHPT1 is higher in breast cancer cells than in normal breast tissue and is higher in ER-positive than in ER-negative breast cancer [17]. The present study showed that survival rates following late recurrence were associated with tCho levels. CHPT1 also plays a role in the invasion of tamoxifen-resistant breast cancer cells, with tamoxifen-resistant LCC2 cells exhibiting greater invasiveness than tamoxifen-sensitive MCF7 cells under CHPT1 expression [17]. Moreover, CHPT1 depletion resulted in stronger suppression of invasion and metastasis by LCC2 cells than by MCF7 cells [17]. ER-positive breast cancer has often been associated with late recurrence [21], with extended hormone therapy administered to many patients with ER-positive breast cancer to reduce the risk of recurrence [22]. Adherence to hormone therapy, however, often decreases after 5 years [23], with many patients not receiving extended hormone therapy. This environment of minimal residual disease (MRD) may function similarly to tamoxifen-resistant breast cancer cells, potentially increasing CHPT1 activity.

---

## [Editor Report · Decision Letter 1]

10 Dec 2024

Prognostic significance of total choline on in-vivo proton MR spectroscopy for prediction of late recurrence in patients with hormone receptor-positive, HER2-negative early breast cancer

PONE-D-24-39699R1

Dear Dr. Kim,

We’re pleased to inform you that your manuscript has been judged scientifically suitable for publication and will be formally accepted for publication once it meets all outstanding technical requirements.

Kind regards,

Seok-Geun Lee, PhD

Academic Editor

PLOS ONE
---

## [Editor Report · Acceptance letter]

13 Dec 2024

PONE-D-24-39699R1 

PLOS ONE

Dear Dr. Kim, 

I'm pleased to inform you that your manuscript has been deemed suitable for publication in PLOS ONE. Congratulations! Your manuscript is now being handed over to our production team.

Kind regards, 

on behalf of

Dr. Seok-Geun Lee 

Academic Editor

PLOS ONE